# The Importance of Genetic Testing in the Differential Diagnosis of Atypical TSC2-PKD1 Contiguous Gene Syndrome—Case Series

**DOI:** 10.3390/children10030420

**Published:** 2023-02-22

**Authors:** Petronella Orosz, Zita Kollák, Ákos Pethő, András Fogarasi, György Reusz, Kinga Hadzsiev, Tamás Szabó

**Affiliations:** 1Bethesda Children’s Hospital, 1146 Budapest, Hungary; 2Department of Pediatrics, Faculty of Medicine, University of Debrecen, 4032 Debrecen, Hungary; 3Department of Internal Medicine and Oncology Clinic, Faculty of Medicine, Semmelweis University, 1083 Budapest, Hungary; 41st Department of Pediatrics, Faculty of Medicine, Semmelweis University, 1083 Budapest, Hungary; 5Department of Medical Genetics, Medical School, Clinical Centre, University of Pécs, 7624 Pécs, Hungary

**Keywords:** tuberous sclerosis, angiomyolipoma, polycystic kidney, contiguous gene syndrome, NTHL1

## Abstract

Background: In clinical practice, the possible diagnosis of tuberous sclerosis or polycystic kidney disease is primarily based on clinical criteria, which can later be verified by genetic testing. But in the case of TSC2/PKD1 contiguous gene syndrome (TSC2/PKD1-CGS), the renal appearance of the disease is more serious. Therefore, early genetic analysis is recommended. Methods: Herein we present the report of four children with TSC2/PKD1-CGS, one involving the NTHL1 gene. We aim to emphasize the importance of genetic testing in this rare syndrome. Results: During the follow-up of tuberous sclerosis and polycystic kidney disease patients, it is essential to reappraise the diagnosis if the clinical symptoms’ appearance or onset time is unusual. Targeted genetic testing is recommended. However, early tumor formation necessitates the extension of genetic analysis. Conclusions: An appropriate evaluation of the phenotype is the cornerstone of diagnosing the rare TSC2/PKD1-CGS with the help of genetic results. In addition, malignant tumors could draw attention to an infrequent large deletion.

## 1. Introduction

Pathogenic mutations cause tuberous sclerosis complex (TSC) in two tumor suppressor genes, TSC1 (9q34) and TSC2 (16p13) [1]. The encoded proteins (hamartin and tuberin) are essential in regulating cell growth and proliferation by inhibiting the mTOR (mammalian target of rapamycin) signaling pathway. Consequently, in the presence of mutations in TSC genes, continuous activation of the mTOR pathway occurs leading to the development of hamartomas in various organs [1]. Common features include benign hamartomas like cardiac rhabdomyoma, subependymal giant cell astrocytoma (SEGA), and renal angiomyolipoma (AML). Malignant tumors, however, are infrequent, and renal cell carcinoma occurs in 2–4% of TSC patients [2].

Renal cysts can occur in about 14–32% of TSC cases [1,3], and two main manifestations are possible. The most common appearance is in classic TSC, where single or multiple small, scarce lesions that are asymptomatic and histologically uniform manifest. A less common cystic manifestation of TSC is associated with autosomal dominant polycystic kidney disease (ADPKD), found in less than 2% of cases [4]. The PKD1 gene, if mutated, is known to cause ADPKD, located on chromosome 16p13.3, which lies immediately adjacent to TSC2. Large genomic deletions disrupt TSC2 and PKD1, causing TSC2/PKD1 contiguous gene syndrome (TSC2/PKD1-CGS) [5,6]. The clinical appearance, in this case, is associated with more severe renal symptoms, such as where the number and size of the cysts increase rapidly up to several cm at an early stage, and the size of the kidneys soon exceeds the age average.

Herein, we describe the case reports of four children in whom the coexistence of tuberous sclerosis and polycystic kidney disease was detected.

## 2. Case Reports

Patient One: At present, he is a 15-year-old boy. Cardiac abnormality was not detected after birth. Epileptic seizures never happened, EEG examinations were always negative, and a cranial MRI revealed subcortical tubers. Postnatal ultrasound (US) revealed small cysts in both kidneys, leading to a diagnosis of polycystic kidney disease. Ultrasound at the age of 3 revealed more than ten cysts in both kidneys, the largest being 13 mm in diameter. The craniocaudal length of the kidneys was above +2SD relative to the age average. The ultrasonography and MRI follow up revealed an abnormal growth in the kidneys with moderately increasing cysts (Figure 1A). At the latest ultrasound, numerous, tiny 1–3 mm angiomyolipoma suspects echogenic foci were detected in both renal parenchyma (Table 1), in association with small hepatic hemangiomas (perhaps AML, 7–8 mm in diameter) and hepatic cysts (20 mm in diameter). His renal function is normal, hypertension was diagnosed at the age of 15, which required amlodipine therapy. To date, he has not received any mTOR-inhibitor treatment. His skin lesions, developed at the age of five (hypopigmented spots, angiofibroma, and Shagreen patch) are typical of tuberous sclerosis. An ophthalmic examination described a small retinal hamartoma in the right eye. Genetic analysis by sequencing the TSC genes did not identify pathogenic mutations. A multiplex ligation-dependent probe amplification (MLPA) confirmed a large deletion of exons 17–42 of the TSC2 gene and the complete deletion of the PKD1 gene in heterozygous form (Figure 2, Appendix A). The genetic analysis of his parents did not verify this deletion, so it seems to be a de novo mutation.

Patient Two: At present, she is a five-year-old girl. An intrauterine ultrasound examination revealed multiplex rhabdomyomas in her heart. Neurologic symptoms started at three months with infantile spasms; hypsarrhythmia was observed by an EEG examination. She currently has complex partial epileptic seizures treated with oral vigabatrin and lamotrigine. A cerebral MRI examination showed subependymal nodules (SENs) and cortical tubers, however, no SEGA was detected. During a postnatal abdominal ultrasound, numerous cysts (3–14 mm in size) were discovered in both kidneys. A close follow up detected fast-growing cysts, as shown in Table 2. At the age of two, angiomyolipoma suspected lesions appeared. An abdominal MRI confirmed the presence of AMLs. After the start of everolimus therapy at the age of two, the growth rate of cysts slowed, and the number and size of detected AMLs remained unchanged (Table 2). Still, a follow up abdominal MRI confirmed an unexpected progression where the largest cyst was more than 40 mm in diameter, and a new cyst (3 mm) developed in the liver (Figure 1A). Her renal function and blood pressure values are in the normal range now. On her forehead, a small hypopigmented patch appeared. The sequencing of the TSC1 and TSC2 genes confirmed no genetic abnormalities. The MLPA analysis of the TSC2 gene revealed that a heterozygous deletion affects exons 14–42 of the TSC2 gene and continues to exons 11–46 of the PKD1 gene (Figure 2, Appendix A). In the genetic study of the parents, no mutation was identified, so the deletion observed in the girl developed de novo.

Patient Three: At present, he is a 13-year-old boy. The postpartum cardiology ultrasound examination confirmed intrauterine findings: large rhabdomyomas in the septum and the front wall of both ventricles, with no outflow obstruction. Rhabdomyomas regressed spontaneously by the age of four. The first epileptic seizure appeared in infancy. A cranial MRI revealed a pervasive manifestation of tuberous sclerosis in the cerebral hemisphere, SENs, an 8 mm SEGA on the left side in the foramen of the Monro area, and numerous cortical tubers. Vigabatrin and then valproate was administered, however, his seizures continued. Therefore, he received everolimus therapy from the age of 2.5. He became seizure-free, and the antiepileptic drug was discontinued later. He underwent neurosurgery three times (at the ages of 9, 10, and 11) due to hydrocephaly caused by SEGA. Unfortunately, he has developed a moderate mental disability. A neonatal abdominal ultrasound examination described a cyst 7 mm in diameter in the right kidney. At six months, 1.5–3.5 cm cysts were revealed in both kidneys, which showed continuous growth in number and size (Table 3). The craniocaudal length of the kidneys was also above the age-matched average. The latest ultrasound and MRI examinations revealed a kidney length percentile of +7 SD and +8 SD above the mean value (Figure 1C). Extrarenal cysts were also discovered in the pancreas and the S4 segment of the liver. Until now, angiomyolipoma could not be confirmed with an MRI or ultrasound. Hypertension developed at 18 months, which necessitated the introduction of antihypertensive therapy. Normal renal function has been preserved since. His dermatology symptoms were a Shagreen spot on the left cheek, hypopigmented spots all over the body, and faded on the right knee since everolimus therapy. A targeted genetic examination with MLPA verified the heterozygous deletion of exons 22–42 of the TSC2 gene and the total deletion of the PKD1 gene (Figure 2, Appendix A). Since the absence of clinical symptoms in his mother and father, we consider it a de novo mutation.

Patient Four: At present, he is a 17-year-old boy. He was born from a complicated pregnancy with multiple cardiac rhabdomyomas diagnosed prenatally, causing aortic valve insufficiency with ventricular outflow tract obstruction. The rhabdomyomas were considered inoperable in newborn age, however, at the age of eight, he underwent corrective surgery for aortic stenosis, and later, he received an artificial valve. Therapy-resistant epileptic seizures in the form of West syndrome started at five months of age. He lives with autism, and he has a moderate mental disability. His polycystic kidneys were recognized in infancy (Table 4). At the age of eight, in the lower pole of the right kidney, a zigzag, thick-walled, echogenic mass with a diameter of 35 mm was visible on abdominal ultrasound, which was described as a ruptured cyst. A follow-up examination revealed a significant progression in the previously detected lesion. Based on an abdominal MRI examination, renal cell cancer was the most likely diagnosis. One week later, he underwent tumor removal surgery. After a right-sided nephrectomy, the histology showed nephroblastoma with a blastemal component, in which there was a minimal, less than 1% epithelial or mesenchymal component. In addition, a remnant of a small angiomyolipoma was recognized in the removed polycystic kidney. After chemotherapy, radiotherapy, and surgical intervention, he was in remission. His renal function remained in the normal range, though modest proteinuria (1 g/day) appeared at the age of 14. Unfortunately, one year later, his eGFR dropped to 64 mL/min/1.73 m^2^, his proteinuria progressed (2.5 g/day), and hypertension developed, requiring ACE-inhibitor therapy. At the age of 16, a common control abdominal MRI presented, aside from numerous cysts, a lesion previously thought to be angiomyolipoma, and it showed a nonspecific appearance and growth (Figure 1D). A close follow up revealed a slow-growing renal cell carcinoma and a metastatic tumor in his liver. Further therapy is in progress. Now, at 17, he is in stage CKD3b (his GFR is 30 mL/min/1.73 m^2^). He also has hypopigmented spots all over his body. Genetic examinations by MLPA revealed that a large heterozygous deletion affected the whole PKD1 gene, the whole TSC2 gene, and, surprisingly, exon one of the NTHL1 gene (Figure 2, Appendix A). Based on his clinically asymptomatic parents, this boy also has a de novo mutation.

## 3. Discussion

TSC2/PKD1-CGS was first described in 1994 by Brook-Carter et al. [7]. In his report, all six presented cases had typical polycystic kidney disease and epileptic seizures in infancy. In addition, five of the six patients had hypertension, and all six had characteristic skin lesions. The initial renal presentation was similar in our cases as the number and diameter of renal cysts were typical (more than ten cysts with a diameter > 2 cm). In all but the first case, fetal rhabdomyoma was the first typical sign of the disease (Appendix A). For this reason, patient one received a diagnosis of polycystic kidney disease at first, while the other three patients received a diagnosis of tuberous sclerosis in infancy. In patient one, the possibility of TSC arose only after the appearance of hypomelanotic macules. In the other three cases, the diagnosis of TSC2/PKD1-CGS was outlined as secondary, based on the rapidly growing cysts.

Although renal cysts are only a minor criterion in diagnosing tuberous sclerosis (Appendix A) [8,9], their presence can be helpful in the differential diagnosis. The diagnosis of TSC2/PKD1-CGS is often based on the typical appearance of renal cysts (diameter exceeding 2 cm at an early stage) and variable coexistence with angiomyolipoma. When comparing TSC patients with TSC/ADPKD patients, the renal phenotype is more severe in the latter, where large cysts predominate over AMLs, and there is a progressive enlargement of the kidneys [10]. Symptomatic kidney cysts are seen in 30–50% of TSC patients with renal manifestations [1]. The rupture of the cysts might cause macroscopic hematuria, however, it is usually not as severe as the bleeding of AMLs. When AMLs are numerous and growing, the risk of renal dysfunction and bleeding increases, especially in lesions larger than 3 cm [1]. Aneurysms can develop in the vessels that supply the angiomyolipoma, which can rupture and cause life-threatening bleeding. In patients one and three, AMLs have not been present in the kidneys for a long time. Patients with large deletions of the TSC2 gene usually show larger AML diameters [11], suggesting that our cases had an unusual phenotype with the deletion of exons 17–42 and 22–42. The typical clinical phenotype of large TSC2 gene deletions was seen in our second case. We emphasize that if imaging results are unusual in TSC or ADPKD, genetic testing may help diagnose TSC2/PKD1-CGS.

The correlation between hypertension and kidney volume has been demonstrated in many studies [12,13,14], indicating the importance of regularly monitoring kidney function and blood pressure, even in asymptomatic patients with stable kidney function. In patients one and three, hypertension was verified despite their retained kidney function. In the fourth case, hypertension developed in parallel with the deterioration of kidney function.

The reported average onset time of malignancies in TSC is 36 years; still, several case reports show that children have an increased risk for malignant tumors [2,15]. Patient four had an atypical occurrence of malignancies. Nephroblastoma was never written with TSC or ADPKD, though it can also occur in the case of WT1 germline and somatic mutations. Unfortunately, we have no data on whether he has a WT1 mutation, however, the coincidence with TSC2/PKD1-CGS would probably be a literary rarity. Some authors assume a relationship between mTOR-pathway activation and Wilms tumor, though the details remain unclear [16]. The second tumor of patient four was renal cell carcinoma, which can be associated with both TSC and ADPKD. But renal cell carcinoma may occur in patients with ADPKD, usually in end-stage renal disease, and it can often be multifocal [17]. This unusual early oncology symptom has highlighted the need for an extension of genetic testing that verified the deletion of exon one in the NTHL1 gene. The NTHL1 gene lies immediately adjacent to TSC2, in a head-to-head position, and encodes a DNA glycosylase protein. An increased risk for colorectal and breast cancer typically characterizes biallelic mutations in NTHL1. Additionally, urothelial and mesothelial carcinomas can occur [18]. Heterozygous mutations in NTHL1 can usually cause benign tumors. A second hit, the “loss of heterozygosity” mutation, could probably explain his tumor, though further examinations are needed. To the best of our knowledge, it is the second case report suggesting that the mutations of these three genes provoked clinical symptoms, and the first describes a large deletion affecting all of the NTHL1-TSC2-PKD1 genes [19].

The first-line treatment of renal lesions in TSC is using mTOR inhibitors. A comprehensive series of studies has shown that everolimus significantly reduces the growth rate of AMLs and cysts [20] as observed in our second patient. Patient three still has no detectable AML, probably due to the everolimus therapy. This treatment has also been used safely in polycystic kidney disease, however, the progression-slowing effect in ADPKD was reported to be less effective [21].

Genetic testing in ADPKD patients with a positive family history and typical symptoms is often not performed, and the diagnosis is based on the clinical picture and inheritance patterns. Tuberous sclerosis can also be diagnosed based on clinical criteria, though genetic tests are widely available. Classic Sanger sequencing or next-generation sequencing (NGS) can identify pathogenic mutations in the exons of TSC1 and TSC2 genes. Large rearrangements (gross deletions/duplications) undetected by sequencing can be confirmed with MLPA. In the case of a suspected contiguous gene syndrome, the genetic examination should start purposefully with an MLPA analysis to detect large deletions. Extensive genetic testing should be considered in the case of early tumor onset.

In tuberous sclerosis, no genotype–phenotype correlations have been established even with the knowledge of the exact mutation since environmental factors and polymorphic variants affecting TSC gene function may modify the clinical appearance of the disease [22,23]. However, phenotypic features are the cornerstones of targeted genetic testing. Obviously, the genetic diagnosis of TSC2/PKD1-CGS predicts the appearance of a more severe phenotype [24], and affected patients need improved attention with assistance to prepare for earlier occurring end-stage renal disease.

## 4. Conclusions

Consistent with previous case reports and reviews, caring for patients with TSC2/PKD1-CGS requires close attention. It can lead to early hypertension and a faster decline of renal function, and end-stage kidney disease develops in the second to third decade of life. Due to the accelerated progression of the disease, more frequent imaging examinations and control tests may become necessary. Differential diagnoses may be challenging in patients whose presentation of the disease is not typical (e.g., the absence of early TSC symptoms with polycystic kidneys). We emphasize that early targeted genetic testing is a priority if TSC2/PKD1-CGS is suspected; extended MLPA or NGS is necessary for affected children with early onset or recurrent malignant tumors.

## Figures and Tables

**Figure 1 children-10-00420-f001:**
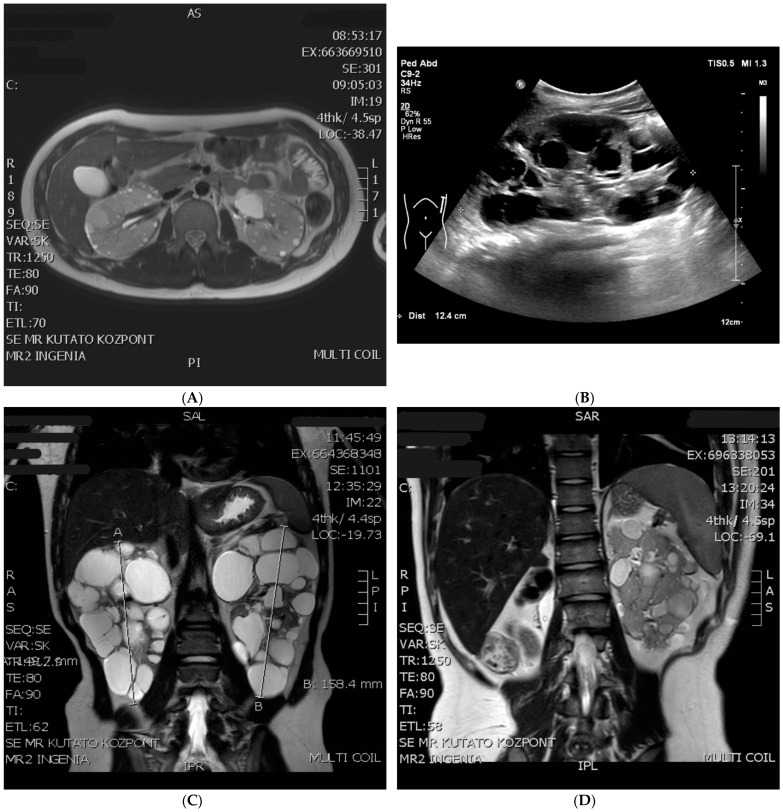
MRI and US pictures of the four patients (**A**): Patient One: radial section of an abdominal MRI at the age of 15, numerous cysts between 1–35 mm in diameter can be seen in both kidneys (**B**): Patient Two: abdominal ultrasound picture of the enormously enlarged (craniocaudal length: 124 mm) left polycystic kidney at the age of 5 (**C**): Patient Three: coronal section of an abdominal MRI with numerous large cysts in both kidneys at the age of 13 (**D**): Patient Four: abdominal MRI after right-sided nephrectomy, the non-specific lesion (renal cell carcinoma) is in the middle third of the kidney.

**Figure 2 children-10-00420-f002:**
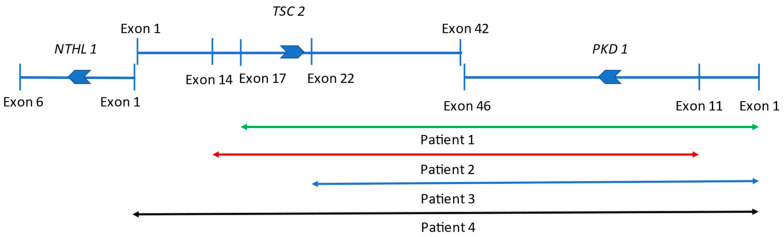
Schematic picture of the affected genes in our contiguous gene syndrome patients. NTHL1, TSC2, and PKD1 lie immediately adjacent to each other. The large deletion that affected TSC2 and PKD1 can disrupt the last exons of TSC2 and PKD1. Joint deletions of NTHL1 and TSC2 are rare, affecting the first exons of both genes. Deleted region in Patient One: green line, extensive rearrangement of Patient Two: red line, mutation of Patient Three: blue line, and copy number variation of Patient Four: black line.

**Table 1 children-10-00420-t001:** Nephrology follow-up imaging studies of Patient One (ultrasound and MRI examination results).

Age at Examination	Kidney Size Right/Left (mm)	Largest Cyst Diameter in the Right/Left Kidney (mm)	Angiomyolipoma Size (mm)
3 years	70/70	13/13	0
5 years	80/80	ND	0
10 years	100/100	25/20	0
15 years	138/136	39/35	3

ND: no data.

**Table 2 children-10-00420-t002:** Nephrology follow-up imaging studies of Patient Two (ultrasound and MRI examination results).

Age at Examination	Kidney Size Right/Left (mm)	Largest Cyst Diameter in the Right/Left Kidney (mm)	Angiomyolipoma Size (mm)
2 weeks	ND	12/14	0
4 months	72/77	18/32	0
9 months	74/74	23/34	0
23 months	86/91	15/26	0
27 months	95/96	18.5/26.5	2–3
34 months	99/103	18/27	2–3
4 years	105/113	18/41	3–6
5 years	113/124	29/25	3–6

Follow-up indicated cysts growing rapidly in size and number. After the initiation of everolimus therapy (thick line), the growth rate slowed. ND: no data.

**Table 3 children-10-00420-t003:** Nephrology follow up imaging studies of Patient Three (ultrasound and MRI examination results).

Age at Examination	Kidney Size Right/Left (mm)	Largest Cyst Diameter in the Right/Left Kidney (mm)	Angiomyolipoma Size (mm)
1 month	ND	7/0	0
6 months	ND	15/35	0
4 years	112/113	33/52	0
6 years	ND	40/50	0
9 years	ND	38/37	0
11 years	152/154	39/47	0
12 years	150/160	50/52	0

Cysts growing rapidly in size and number; the growth rate decreased after the initiation of everolimus therapy (thick line) ND: no data.

**Table 4 children-10-00420-t004:** Nephrology follow up imaging studies of Patient Four (ultrasound and MRI examination results).

Age at Examination	Kidney Size Right/Left (mm)	Largest Cyst Diameter in the Right/Left Kidney (mm)	Angiomyolipoma Size (mm)
5 months	120/120	ND	0
2 years	130/130	48/15	0
4 years	130/130	20/20	0
6 years	140/140	41/30	0
8 years	140/140	42/33	0
9 years	-/145	-/28	0
11 years	-/148	-/31	0
13 years	-/170	-/35	0
15 years	-/170	-/47	2–3
16 years	-/175	-/50	2–3
17 years	-/180	-/50	2–3

ND: no data.

## Data Availability

Presented data are available on request from the corresponding author. However, the data are not public due to privacy reasons.

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
