# Peer review of "The Importance of Genetic Testing in the Differential Diagnosis of Atypical TSC2-PKD1 Contiguous Gene Syndrome—Case Series"

_children, 2023, doi:10.3390/children10030420_

Round 1
Reviewer 1 Report
This is a case report about the atypical TSC2-PKD1 contiguous gene syndrome. A total of 4 cases was presented. In general, the manuscript contains many important information for the clinical field in the area. Minor comments to improve presentation rigors are as follow.
1. The gene sequences from the genetic testings should be presented as supplemental data. While Fig.2 is good, it does not have enough resolution.
2. While we agree with the authors that targeted genetic testing is needed, the authors need to provide strong rationales as to why genetic testing is recommended. The counter argument is that if there is no current treatment for the syndrome, then how the genetic testing helps the patients.
3. Minor: Because the studies lasted for more than 17 years, please provide all IRB numbers approved to carry out these studies.
Author Response
To Rewiever #1.
- The gene sequences from the genetic testings should be presented as supplemental data. While Fig.2 is good, it does not have enough resolution.
Answer: Thank you for your opinion. With Figure 2. we would like to give only a schematic picture of the mutations to help visualize the localization of large deletions. For better clarification, we made a supplementary table about the genetic test results of the four patients.
- While we agree with the authors that targeted genetic testing is needed, the authors need to provide strong rationales as to why genetic testing is recommended. The counter argument is that if there is no current treatment for the syndrome, then how the genetic testing helps the patients.
Answer: Thank you for your comment. All affected patients have the right to know whether they will develop the end-stage renal disease at the age of 20-30, or only later, in their 50s. Unfortunately, we are not able to cure them, but with the knowledge of the genetic test results, we can increase attention, more frequent control examinations could happen, and we can prepare the patients in time for the potential kidney transplantation procedure and renal replacement therapy. These are the most important goals of regular nephrology care. Other authors concluded, too, that the genetic diagnosis of TSC2/PKD1-CGS predicts the appearance of a more severe phenotype; we mention it in the corrected manuscript.
- Minor: Because the studies lasted for more than 17 years, please provide all IRB numbers approved to carry out these studies.
Answer: Thank you for your comment. In our manuscript, we describe about the retrospectively collected cases of the patients we care for. The follow-up was not conducted as a study or clinical trial. The genetic test happened in the context of comprehensive and precise patient care, with parental consent after appropriate information in all cases. The parents gave written consent also to the publication of the data in the manuscript. During our cohort, no experimental drugs nor experimental investigation were used. However, our working group has the permission of the local ethics committee to conduct a non-interventional clinical trial among tuberous sclerosis patients. (Protocol number: BTH2020/01)
Reviewer 2 Report
In the current case report, authors have reassessed the importance of genetic testing in TSC2/PKD1 contiguous gene syndrome. Accordingly, authors have evaluated the phenotype of four children diagnosed with TSC2/PKD1-CGS. Subsequently, multiplex ligation dependent probe amplification was employed for genetic analysis and confirmed de novo heterozygous genotypes in the study subjects. As mentioned by authors, progression of tuberous sclerosis doesn’t show a genotype-phenotype correlations while polycystic kidney disease correlates with a dose dependent expression of PKD1 gene. Priority genetic testing certainly assists in treating TSC2/PKD1-CGS, especially in children with atypical progression. Presented findings justifies this and profoundly contributes to the existing literature. Few minor corrections need to be addressed as follows
Abbreviations should be elaborated at their first mention: as in line 58 “US”
Occasional grammatical errors
The arrowhead presenting the orientation of PKD1 gene should be in reverse in Fig 2.
Author Response
To Rewiever #2.
Abbreviations should be elaborated at their first mention: as in line 58 “US”
Answer: Thank you for your comment; we corrected the explanation in line 58.
Occasional grammatical errors
Answer: Thank you for your comment. We corrected grammatical errors.
The arrowhead presenting the orientation of the PKD1 gene should be in reverse in Fig 2.
Answer: Thank you very much for your comment; we have corrected the figure.